# SOME CONSIDERATIONS ON LEARNING TO EXPLORE VIA META-REINFORCEMENT LEARNING

## ABSTRACT

We consider the problem of exploration in meta reinforcement learning. Two new meta reinforcement learning algorithms are suggested: E-MAML and E-RL$^2$. Results are presented on a novel environment we call 'Krazy World' and a set of maze environments. We show E-MAML and E-RL$^2$ deliver better performance on tasks where exploration is important.

## 1 INTRODUCTION

Supervised learning algorithms typically have their accuracy measured against some held-out test set which did not appear at training time. A supervised learning model generalizes well if it maintains its accuracy under data that has non-trivial dissimilarity from the training data. This approach to evaluating algorithms based on their generalization ability contrasts the approach in reinforcement learning (RL), wherein there is usually no distinction between training and testing environments. Instead, an agent is trained on one environment, and results are reported on this same environment. Most RL algorithms thus favor mastery over generalization.

Meta RL is the suggestion that RL should take generalization seriously. In Meta RL, agents are evaluated on their ability to quickly master new environments at test time. Thus, a meta RL agent must not learn how to master the environments it is given, but rather it must *learn how to learn* so that it can quickly train at test time. Recent advances in meta RL have introduced algorithms that are capable of generating large policy improvements at test time with minimal sample complexity requirements Finn et al. (2017); Duan et al. (2016); Wang et al. (2016).

One key question for meta RL that has been inadequately considered by Finn et al. (2017); Duan et al. (2016) is that of exploration (see Kompella et al. (2002) for an overview of exploration in this context). A crucial step in learning to solve many environments is an initial period of exploration and system identification. Furthermore, we know that real-life agents become better at this exploratory phase with practice. Consider, for example, an individual playing an entirely new video game. They will first need to identify the objective of the game and its mechanics. Further, we would expect that individuals who have played many video games would have a significantly easier time learning new games. Similarly, we would expect a good meta RL agent to become more efficient at this exploratory period. Unfortunately, we have found existing algorithms deficient in this area. We hypothesize that this failure can at least partially be attributed to the tendency of existing algorithms to do greedy, local optimization at each step of meta-training, as discussed further below.

To address the problem of exploration in meta RL, we introduce two new algorithms: E-MAML and E-RL$^2$. It should come as no surprise that these algorithms are similar to their respective namesakes MAML and RL$^2$. The algorithms are derived by reformulating the underlying meta-learning objective to account for the impact of initial sampling on future (post-meta-updated) returns. We show that our algorithms achieve better results than MAML and RL$^2$ on two environments: a Krazy World environment we developed to benchmark meta RL, and a set of maze environments. In the appendix to this work, we consider a more general form of our E-MAML derivation. This more general derivation suggests several promising directions for future work on exploration in meta-learning and highlights the novelty of our contributions.

## 2 RELATED WORK

Recently, a flurry of new work in Deep Reinforcement Learning has provided the foundations for tackling RL problems that were previously thought intractable. This work includes: 1) Mnih et al. (2015; 2016); Koutnk et al. (2013), which allow for discrete control in complex environments directly from raw images. 2) Schulman et al. (2015); Mnih et al. (2016); Schulman et al. (2017); Lillicrap et al. (2015), which have allowed for high-dimensional continuous control in complex environments from raw state information.

Although these algorithms offer impressive capabilities, they still often falter when faced with environments where exploration presents a significant challenge. This is not surprising, as there exists a large body of RL work addressing the exploration problem (Kearns & Singh, 2002; Brafman & Tennenholtz, 2002; Kolter & Ng, 2009). In practice, these methods are often not used due to difficulties with high-dimensional observations, difficulty in implementation on arbitrary domains, and lack of promising results (though recent work such as Schmidhuber (2015a) has seen progress in this regard). This resulted in most deep RL work utilizing epsilon greedy exploration (Mnih et al., 2015), or perhaps a simple scheme like Boltzmann exploration (Carmel & Markovitch, 1999). As a result of these shortcomings, a variety of new approaches to exploration in deep RL have recently been suggested (Tang et al., 2016; Houthooft et al., 2016; Stadie et al., 2015; Osband et al., 2016; Bellemare et al., 2016; Ostrovski et al., 2017). Further, there has been extensive research on the idea of artificial curiosity. For example Ngo et al. (2012); Graziano et al. (2011); Schmidhuber (1991; 2015b); Storck et al. (1995); Sun et al. (2011) all deal with teaching agents to generate a curiosity signal that aids in exploration. The work Kompella et al. (2002) even considers the problem of curiosity in the context of meta-RL via Success Story Algorithms (SSA). These algorithms use dynamic programming to explicitly partition experience into checkpoints and serve as a precursor to MAML and RL$^2$, which are less focused on explicitly partitioning episodes and more focused on handling high dimensional input Schmidhuber et al. (1997). In spite of the numerous cited efforts in the above paragraph, the problem of exploration in RL remains difficult.

This paper will consider the problem of exploration in meta RL. However, it is important to note that many of the problems in meta RL can alternatively be addressed with the field of *hierarchical reinforcement learning*. In hierarchical RL, a major focus is on learning primitives that can be reused and strung together. These primitives will frequently enable better exploration, since they'll often relate to better coverage over state visitation frequencies. Recent work in this direction includes (Vezhnevets et al., 2017; Bacon & Precup, 2015; Tessler et al., 2016; Rusu et al., 2016; Barto & Mahadevan, 2003; Wiering & Schmidhuber, 1997). The primary reason we consider meta learning over hierarchical RL is that we find hierarchical RL tends to focus more on defining specific architectures that should lead to hierarchical behavior, whereas meta learning instead attempts to directly optimize for these behaviors.

As for meta RL itself, the literature is spread out and goes by many different names. There exist relevant literature on life-long learning, learning to learn, continual learning, and multi-task learning. Consider for instance Schmidhuber (2006; 1987). These papers deal with self modifying learning machines or genetic programs. In particular, they propose that if you have a genetic program that itself modifies the underlying genetic program, you will obtain a meta-GP approach (see appendix B for some clarification of this approach to the the algorithms presented in this paper). These methods describe an extremely broad class of algorithms which could be said to encompass most meta-learning approaches with the correct interpretation. In addition to these references, we encourage the reviewer to look at the review articles Silver et al. (2013); Taylor & Stone (2009); Thrun (1996) and their citations. Most attempts to address exploration in these contexts has focused on curiosity or on a free learning phrase during training. Curiosity is addressed above, and is quite different from this work because it focuses on defining an intrinsic motivation signal, whereas we only consider better utilization of the existing reward signal. This paper's methods share more similarities with free learning phases, though our algorithm makes a more explicit connection between this phase and the more exploitative phase of training (directly incorporating this information into the gradient updates).

This paper will seek to capture insights from this prior literature and apply them to recently developed algorithms. In particular, we will consider considering E-MAML and E-RL$^2$, which build upon the existing meta learning algorithms MAML (Finn et al., 2017) and RL$^2$ (Duan et al., 2016).

In MAML, tasks are sampled and a policy gradient update is computed for each task with respect to a fixed initial set of parameters. Subsequently, a meta update is performed where a gradient step is taken that moves the initial parameter in a direction that would have maximally benefited the average return over all of the sub-updates. Results are presented on an environment wherein a simulated ant must determine its goal direction, which changes each time a new task is sampled. In $RL^2$, an RNN is made to ingest multiple episodes from many different MDPs and then perform a policy gradient update through the entire temporal span of the RNN. The hope is that the RNN will learn a faster RL algorithm in its memory weights.

## 3 REINFORCEMENT LEARNING NOTATION

Let $M = (\mathcal{S}, \mathcal{A}, \mathcal{P}, r, \rho_0, \gamma, T)$ represent a discrete-time finite-horizon discounted Markov decision process (MDP). The elements of $M$ have the following definitions: $\mathcal{S}$ is a state set, $\mathcal{A}$ an action set, $\mathcal{P} : \mathcal{S} \times \mathcal{A} \times \mathcal{S} \to \mathbb{R}_+$ a transition probability distribution, $r : \mathcal{S} \times \mathcal{A} \to \mathbb{R}$ a reward function, $\rho_0 : \mathcal{S} \to \mathbb{R}_+$ an initial state distribution, $\gamma \in [0, 1]$ a discount factor, and $T$ the horizon. We will sometimes speak of $M$ having a loss function $\mathcal{L}$ rather than reward function $r$. All we mean here is that $\mathcal{L}(s) = -r(s)$

In a classical reinforcement learning setting, we optimize to obtain a set of parameters $\theta$ which maximize the expected discounted return under the policy $\pi_\theta : \mathcal{S} \times \mathcal{A} \to \mathbb{R}_+$. That is, we optimize to obtain $\theta$ that maximizes $\eta(\pi_\theta) = \mathbb{E}_{\pi_\theta}[\sum_{t=0}^{T} \gamma^t r(s_t)]$, where $s_0 \sim \rho_0(s_0)$, $a_t \sim \pi_\theta(a_t|s_t)$, and $s_{t+1} \sim \mathcal{P}(s_{t+1}|s_t, a_t)$.

## 4 PROBLEM FORMULATION AND ALGORITHMS

### 4.1 THE META REINFORCEMENT LEARNING OBJECTIVE

In meta reinforcement learning, we consider a family of MDPs $\mathcal{M} = \{M_i\}_{i=1}^{N}$ which comprise a distribution of tasks. The goal of meta RL is to find a policy $\pi_\theta$ and paired update method $U$ such that, given $M_i \sim \mathcal{M}$, we have $\pi_{U(\theta)}$ solves $M_i$ quickly. The word quickly is important here, so let us elaborate. By quickly, we mean orders of magnitude more sample efficient than simply solving $M_i$ with policy gradient or value iteration methods. In fact, an ideal meta RL algorithm would solve these tasks by collecting 1-10 trajectories from the environment for tasks where policy gradients require 100,000 or more trajectories. The thinking here goes that if an algorithm can solve a problem with so few samples, then it might be 'learning to learn.' That is, the agent is not learning how to master a given task but rather how to quickly master new tasks. We can write this objective cleanly as

$$\min_\theta \sum_{M_i} \mathbb{E}_{(\pi_{U(\theta)})}[\mathcal{L}_{M_i}] \tag{1}$$

Note that this objective is similar to the one that appears in MAML Finn et al. (2017), which we will discuss further below. In MAML, $U$ is chosen to be the stochastic gradient descent.

### 4.2 E-MAML

We can expand the expectation from (1) into the integral

$$\int R(\tau) \pi_{U(\theta)}(\tau) \mathrm{d}\tau \tag{2}$$

The objective (1) can be optimized by taking a derivative of this integral with respect to $\theta$ and carrying out a standard REINFORCE style analysis to obtain a tractable expression for the gradient Williams (1992).

One issue with this direct optimization is that it will not account for the impact of the original sampling distribution $\pi_\theta$ on the future rewards $R(\tau), \tau \sim \pi_{U(\theta)}$. We would like to account for this, because it would allow us to weight the initial samples $\bar{\tau} \sim \pi_\theta$ by the expected future returns we expect after the sampling update $R(\tau)$. More importantly, this extra-term reinforces particular

explorative trajectories if the updated agent learn well in the end. On the contrary when this term is abscent, good exploratory behaviors will not be encouraged.

Including this dependency can be done by writing the modified expectation as

$$\iint R(\tau)\pi_{U(\theta)}(\tau)\pi_\theta(\bar{\tau})\mathrm{d}\bar{\tau}\mathrm{d}\tau \tag{3}$$

Note that one could certainly argue that this is the correct form of (2), and that the failure to include the dependency above makes (2) incomplete. We choose to view both as valid approaches, with (3) simply being the more exploratory version of the objective for reasons discussed above.

In any case, we find ourselves wishing to find a tractable expression for the gradient of (3). This can be done quite smoothly by applying the product rule under the integral sign and going through the REINFORCE style derivation twice to arrive at a two term expression, one of which encourages exploitation and one of which encourages exploration.

$$\frac{\partial}{\partial\theta}\iint R(\tau)\pi_{U(\theta)}(\tau)\pi_\theta(\bar{\tau})\mathrm{d}\bar{\tau}\mathrm{d}\tau$$
$$= \iint R(\tau)\left[\pi_\theta(\bar{\tau})\frac{\partial}{\partial\theta}\pi_{U(\theta)}(\tau) + \pi_{U(\theta)}(\tau)\frac{\partial}{\partial\theta}\pi_\theta(\bar{\tau})\right]\mathrm{d}\bar{\tau}\mathrm{d}\tau$$
$$\approx \frac{1}{T}\sum_{i=1}^T R(\tau^i)\frac{\partial}{\partial\theta}\log\pi_{U(\theta)}(\tau^i) + \frac{1}{T}\sum_{i=1}^T R(\tau^i)\frac{\partial}{\partial\theta}\log\pi_\theta(\bar{\tau}^i) \Bigg|_{\substack{\bar{\tau}^i \sim \pi_\theta \\ \tau^i \sim \pi_{U(\theta)}}} \tag{4}$$

Note that, if we only consider the term on the left, we arrive at the original MAML algorithm Finn et al. (2017). This term encourages the agent to take update steps $U$ that achieve good final rewards. It is exploitative. The second term encourages the agent to take actions such that the eventual meta-update yields good rewards (crucially, it does not try and exploit the reward signal under its own trajectory $\bar{\tau}$). This term allows the policy to be more exploratory, as it will attempt to deliver the maximal amount of information useful for the future rewards $R(\tau)$ without worrying about its own rewards $R(\bar{\tau})$. Since this algorithm augments MAML by adding in an exploratory term, we call it E-MAML. For the experiments presented in this paper, we will assume that $U$ utilized in MAML and E-MAML is stochastic gradient descent. However, we direct the reader to the appendix for an insightful discussion on the choice of $U$.

## 4.3   E-RL$^2$

RL$^2$ optimizes (1) by feeding multiple rollouts from multiple different MDPs into an RNN. The hope is that the RNN hidden state update $h_t = C(x_t, h_{t-1})$, will learn to act as the update function $U$. Then, performing a policy gradient update on the RNN will correspond to optimizing the meta objective (1).

We can write the RL$^2$ update rule more explicitly in the following way. Suppose $L$ represents an RNN. Let $\mathrm{Env}_k(a)$ be a function that takes an action, uses it to interact with the MDP representing task $k$, and returns the next observation $o$[1], reward $r$, and a termination flag $d$. Then we have

$$x_t = [o_{t-1}, a_{t-1}, r_{t-1}, d_{t-1}] \tag{5}$$
$$[a_t, h_{t+1}] = L(h_t, x_t) \tag{6}$$
$$[o_t, r_t, d_t] = \mathrm{Env}_k(a_t) \tag{7}$$

To train this RNN, we sample $N$ MDPs from $\mathcal{M}$ and obtain $k$ rollouts for each MDP by running the MDP through the RNN as above. We then compute a policy gradient update to move the RNN parameters in a direction which maximizes the returns over the $k$ trials performed for each MDP.

To make this algorithm more exploratory, we can take inspiration from the insights shone by E-MAML. For each MDP $M_i$, we will sample $p$ exploratory rollouts and $k-p$ non-exploratory rollouts. During an exploratory rollout, the forward pass through the RNN will receive all information.

---

[1]RL$^2$ works well with POMDP's because the RNN is good at system-identification. This is the reason why we chose to use $o$ as in "observation" instead of $s$ for "state" in this formulation.

However, during the backwards pass, the rewards contributed during exploratory episodes will be set to zero. For example, if there is one exploratory episode and one exploitative episode, then we would proceed as follows. During the forward pass, the RNN will receive all information regarding rewards for both episodes. During the backwards pass, the returns will be computed as

$$R(x_i) = \sum_{j=i}^{T} \gamma^j r_j \cdot \chi_E(x_j) \tag{8}$$

Where $\chi_E$ is an indicator that returns 0 if the episode is exploratory and 1 otherwise. This return, and not the standard sum of discounted returns, is what is used to compute the policy gradient. The hope is that zeroing the return contributions from exploratory episodes will encourage the RNN to treat its initial rollouts as exploratory, taking actions which may not lead to immediate rewards but rather to the RNN hidden weights performing better system identification, which will lead to higher rewards in later episodes.

## 5 EXPERIMENTS

### 5.1 KRAZY WORLD ENVIRONMENT

To test the ability of meta RL algorithms to explore, we introduce a new environment known as Krazy World.

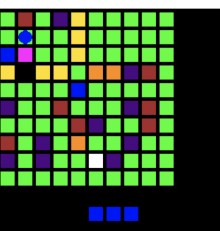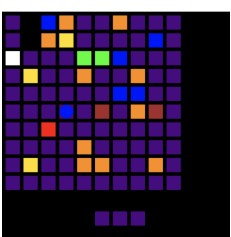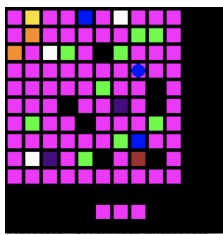

Figure 1: Three example worlds drawn from the task distribution. The agent must first perform an exploratory stage of system identification before exploiting. For example, in the leftmost grid the agent must first identify that the orange squares give +1 reward and the blue squares replenish its limited supply of energy. Further, it will need to identify that the gold squares block progress and the black square can only be passed by picking up the pink key. The agent may also want to identify that the brown squares will kill it and that it will slide over the purple squares. The other center and right worlds show these assignments will change and need to be re-identified every time a new task is drawn.

This environment has the following features:

1. **8 different tile types**: *Goal squares* provide +1 reward when retrieved. The agent reaching the goal does not cause the episode to terminate, and there can be multiple goals. *Ice squares* will be skipped over in the direction the agent is transversing. *Death squares* will kill the agent and end the episode. *Wall squares* act as a wall, impeding the agent's movement. *Lock squares* can only be passed once the agent has collected the corresponding key from a key square. *Teleporter squares* transport the agent to a different teleporter square on the map. *Energy squares* provide the agent with additional energy. If the agent runs out of energy, it can no longer move. The agent proceeds normally across **normal squares**.

2. **Ability to randomly swap color palette**: The color palette for the grid can be randomly permuted, changing the color that corresponds to each of the different tile types. The agent will thus have to identify the new system to achieve a high score. Note that in representations of the gird wherein basis vectors are used rather than images to describe the state space, each basis vector will correspond to a tile type, and permuting the colors will correspond to permuting the types of tiles these basis vectors represent. We prefer to use the basis vector representation in our experiments, as it is more sample efficient.

3. **Ability to randomly swap dynamics**: The game's dynamics can be altered. The most naive alteration simply permutes the player's inputs and corresponding actions (issuing the command for down moves the player up etc). More complex dynamics alterations allow the agent to move multiple steps at a time in arbitrary directions, making the movement more akin to that of chess pieces.

4. **Local or Global Observations**: The agent's observation space can be set to some fixed number of squares around the agent, the squares in front of the agent, or the entire grid. Observations can be given as images or as a grid of basis vectors. For the case of basis vectors, each element of the grid is embedded as a basis vector that corresponds to that tile type. These embeddings are then concatenated together to form the observation proper. We will use local observations.

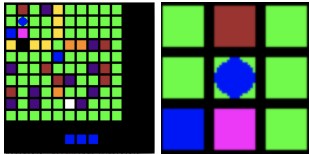

Figure 2: A comparison of local and global observations for the Krazy World environment. In the local mode, the agent only views a $3 \times 3$ grid centered about itself. In global mode, the agent views the entire environment.

A successful meta learning agent will first need to explore the environment, identifying the different tile types, color palette, and dynamics. A meta learning algorithm that cannot learn to incorporate exploration into its quick updating strategy will ultimately have insufficient data to navigate the game. We see this empirically in our experiments below.

## 5.2 MAZES

A collection of maze environments. The agent is placed at a random square within the maze and must learn to navigate the twists and turns to reach the goal square. A good exploratory agent will spend some time learning the maze's layout in a way that minimizes repetition of future mistakes. The mazes are not rendered, and consequently this task is done with state space only. The mazes are $20 \times 20$ squares large.

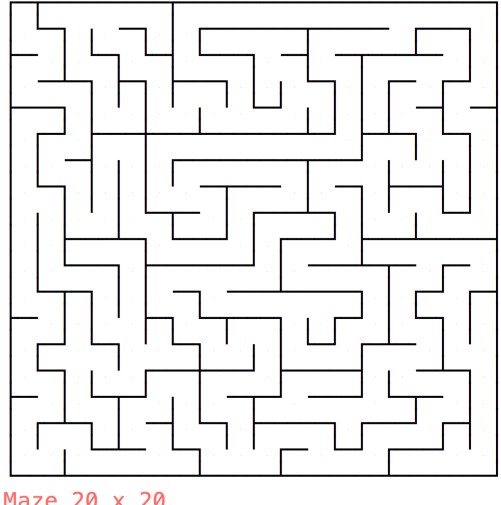

Maze 20 x 20

Figure 3: One example maze environment rendered in human readable format. The agent attempts to find a goal within the maze.

### 5.3 RESULTS

In this section we present the following experimental results.

1. Learning curves on Krazy World and mazes.

2. The gap between the agent's initial performance on new environments and its performance after updating. A good meta learning algorithm will have a large gap after updating. A standard RL algorithm will have virtually no gap after only one update.

3. Three heuristic metrics for how well the algorithms explore.

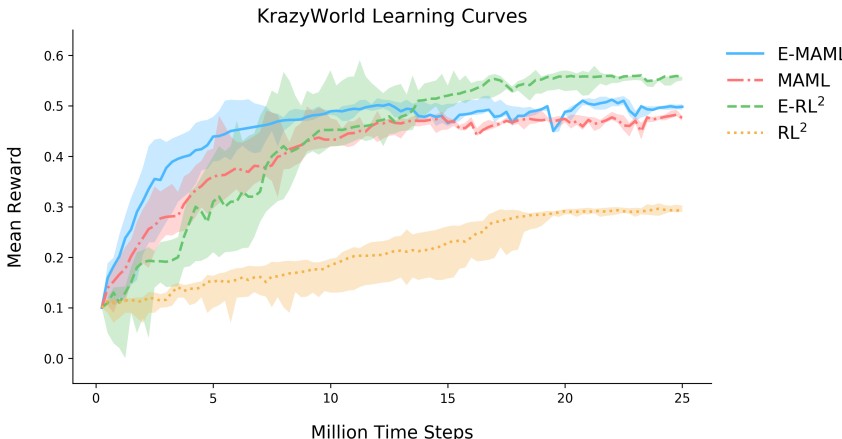

Figure 4: Meta learning curves on Krazy World. We see that -RL$^2$ is at times closest to the optimal solution of 0.6 (obtained via value iteration), but has the highest initial variance. E-MAML converges faster than MAML, although both algorithms do manage to converge. RL$^2$ has relatively poor performance and high variance. A random agent achieves a score of around 0.05 on this task.

For both Krazy World and mazes, training proceeds in the following way. First, we initialize 32 training environments and 64 test environments. Every initialized environment has a different seed. Next, we initialize our chosen meta-RL algorithm by uniformly sampling hyper-parameters from predefined ranges. Data is then collected from all 32 training environments. The meta-RL algorithm then uses this data to make a meta-update, as detailed in the algorithm section of this paper. The meta-RL algorithm is then allowed to do one training step on the 64 test environments to see how fast it can train at test time. These test environment results are recorded, 32 new tasks are sampled from the training environments, and data collection begins again. For MAML and E-MAML, training at test time means performing one VPG update at test time (see the appendix for evidence that taking only one gradient step is sufficient). For RL$^2$ and E-RL$^2$, this means running three exploratory episodes to allow the RNN memory weights time to update and then reporting the loss on the fourth and fifth episodes. For both algorithms, meta-updates are calculated with PPO Schulman et al. (2017). The entire process from the above paragraph is repeated from the beginning 64 times and the results are averaged to provide the final learning curves featured in this paper.

When plotting learning curves in Figure 4 and Figure 5, the Y axis is the reward obtained after training at test time on a set of 64 held-out test environments. The X axis is the total number of environmental time-steps the algorithm has used for training. Every time the environment advances forward by one step, this count increments by one. This is done to keep the time-scale consistent across meta-learning curves.

For Krazy World, learning curves are presented in Figure 4. We see that E-MAML and E-RL$^2$ have the best final performance. E-MAML has the steepest initial gains for the first 10 million time-steps. Since meta-learning algorithms are often very expensive, having a steep initial ascent is quite valuable. Around 14 million training steps, E-RL$^2$ passes E-MAML for the best performance. By 25 million time-steps, E-RL$^2$ is somewhat close to the optimal solution (obtained via value iteration). However, it appears to have converged below this value. MAML delivers comparable

final performance to E-MAML, suggesting that perhaps MAML also learns to explore sufficiently well on this task. However, it takes it much longer to obtain this level of performance. Finally, $RL^2$ has comparatively poor performance on this task and very high variance. When we manually examined the $RL^2$ trajectories to figure out why, we saw the agent consistently finding a single goal square and then refusing to explore any further. The exploration metrics presented below seem consistent with this finding.

Learning curves for mazes are presented in Figure 5. Here, the story is different than Krazy World. $RL^2$ and E-$RL^2$ both perform better than MAML and E-MAML. We suspect the reason for this is that RNNs are able to leverage memory, which is more important in mazes than in Krazy World. This environment carries a penalty for hitting the wall, which MAML and E-MAML discover quickly. However, it takes E-$RL^2$ and $RL^2$ much longer to discover this penalty, resulting in worse performance at the beginning of training. MAML delivers worse final performance and typically only learns how to avoid hitting the wall. $RL^2$ and E-MAML sporadically solve mazes. E-$RL^2$ manages to solve many of the mazes.

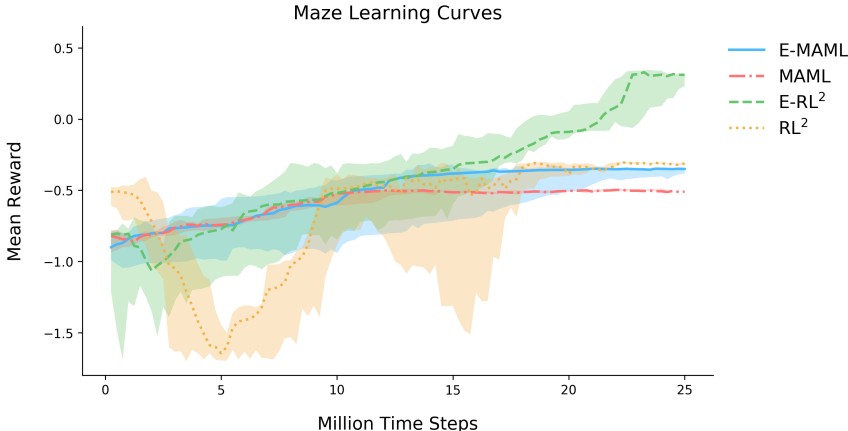

Figure 5: Meta learning curves on mazes. The environment gives the agent a penalty for hitting the wall. Interestingly, E-$RL^2$ and $RL^2$ take much longer to learn how to avoid the wall. However, they also deliver better final performance. E-$RL^2$ learns faster than $RL^2$ and offers superior final performance. Since MAML and E-MAML utilize MLPs instead of RNNs and have no memory, their worse returns are not surprising. The optimal return for this task would be around 1.1 (obtained via value iteration) and a random agent would achieve a return of around -2.2 (mostly from hitting the wall). Note that this figure appears somewhat busy due to having all four algorithms on one graph. Figure 6 shows each curve in isolation, making it easier to discern their individual characteristics.

When examining meta learning algorithms, one important metric is the update size after one learning episode. Our meta learning algorithms should have a large gap between their initial policy, which is largely exploratory, and their updated policy, which should often solve the problem entirely. For MAML, we look at the gap between the initial policy and the policy after one policy gradient step (see the appendix for information on further gradient steps). For $RL^2$, we look at the results after three exploratory episodes, which give the RNN hidden state $h$ sufficient time to update. Note that three is the number of exploratory episodes we used during training as well. This metric shares similarities with the Jump Start metric considered in prior litearture Taylor & Stone (2009). These gaps are presented in 6.

Finally, in Figure 7 we consider three heuristic metrics for measuring the exploratory ability of the meta-learners. First, we consider the fraction of tile types visited by the agent at test time. Our intuition is that good exploratory agents will visit and identify many different tile types as this should aid in optimally solving the game. Second, we consider the number of times an agent visits a death tile at test time. Agents that are efficient at exploring should visit this tile type exactly once and then avoid it. We often find that more naive agents will run into these tiles repeatedly, causing their death many times over and a sense of pity in any onlookers. Finally, we look at how many goals the agent reaches. We see that $RL^2$ tends to visit fewer goals and instead tend to stick with a strategy

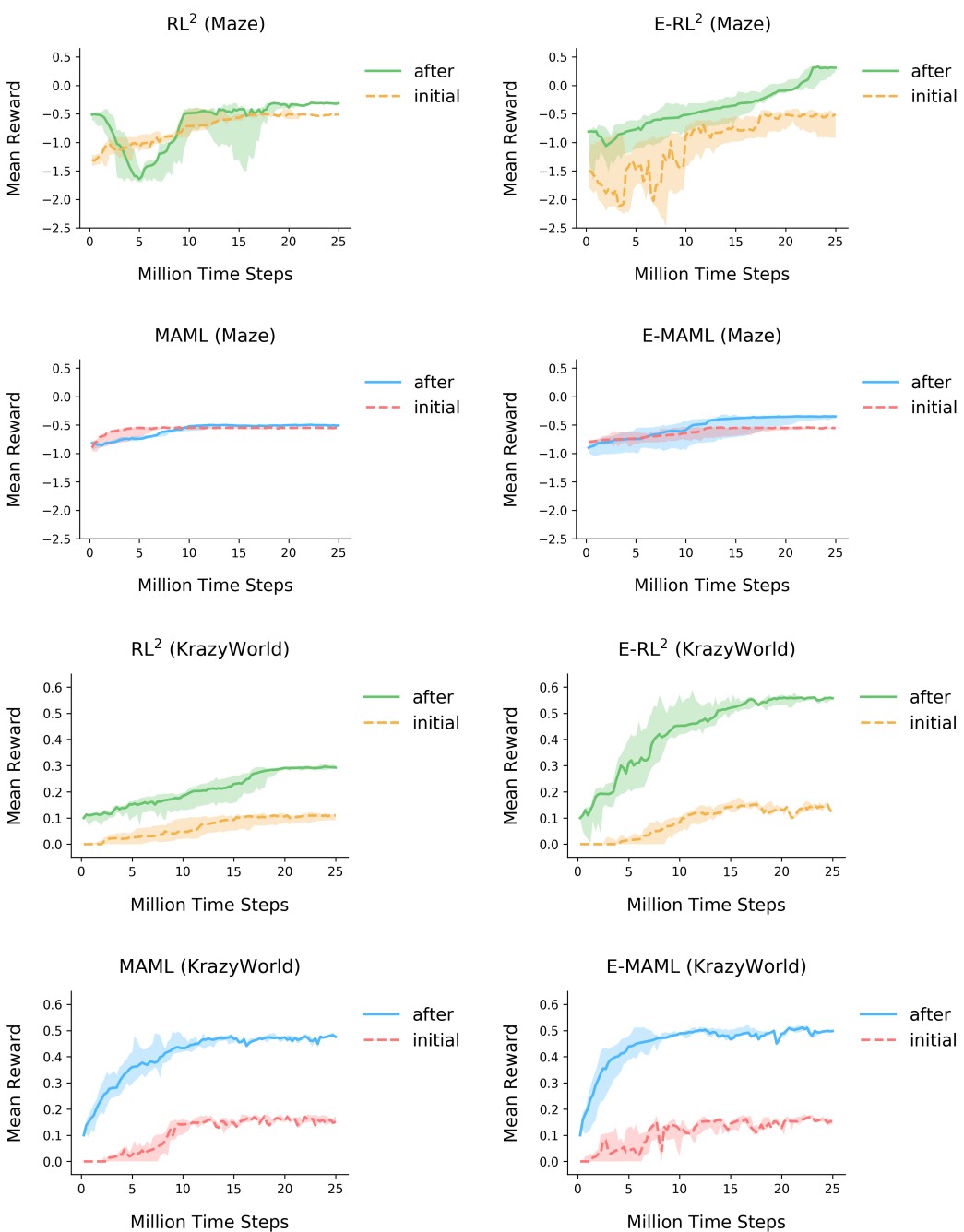

Figure 6: Looking at the gap between initial performance and performance after one update for $RL^2$, $E$-$RL^2$, MAML, and E-MAML. All algorithms show some level of improvement after one update, which suggests the meta-learning is working. On Krazy World, the gap between the initial policy performance and updated policy performance is fairly large for all algorithms. Interestingly, the initial performance of all algorithms on gridworld is similar, converging to values around 0.1. However, E-MAML and $RL^2$ tend to have the highest variance. Meanwhile, for mazes, we see the gap between initial and update policy returns for MAML and E-MAML is unfortunately small. Perhaps this is because these algorithms do not have a memory, which makes the maze task more difficult. For $RL^2$ and $E$-$RL^2$ the gap is bigger, with $E$-$RL^2$ delivering the best improvement.

of finding one goal and exploiting it. Overall, we see that the exploratory algorithms achieve better performance under these metrics.

# 6 CONCLUSION

We considered the problem of exploration in meta reinforcement learning. Two new algorithms were derived and their properties were analyzed. In particular, we showed that these algorithms tend to learn more quickly and explore more efficiently than existing algorithms. It is likely that future work in this area will focus on meta-learning a curiosity signal which is robust and transfers across tasks. Perhaps this will enable meta agents which learn to explore rather than being forced to explore by mathematical trickery in their objectives. See appendix B for more explicit discussion on possible future directions.

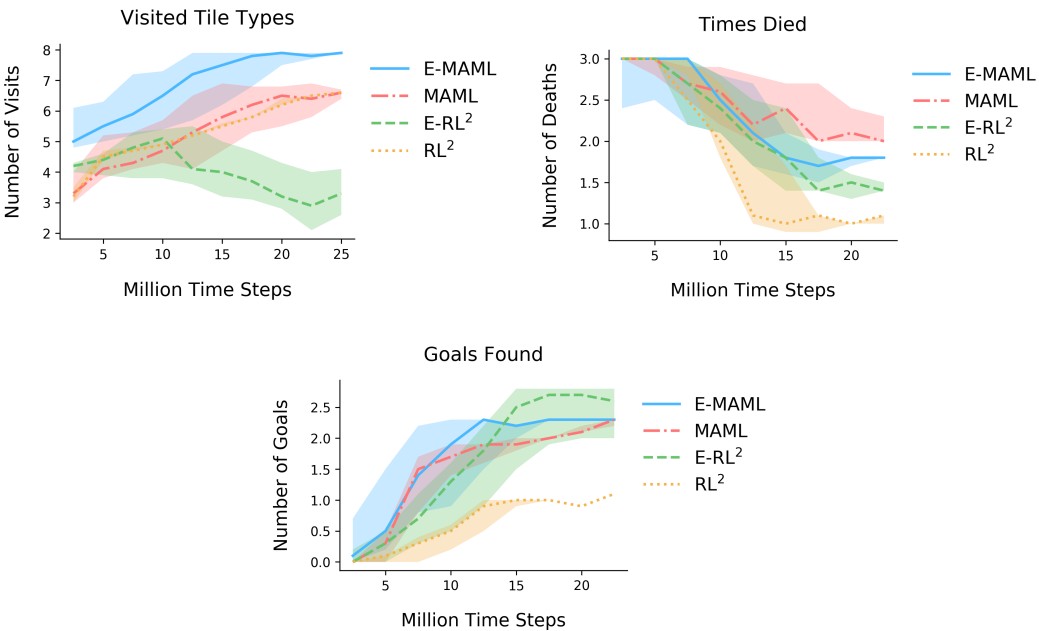

Figure 7: Three heuristic metrics for exploration on Krazy World: Fraction of tile types visited during test time, number of times killed at a death square during test time, and number of goal squares visited. We see that E-MAML is consistently the most diligent algorithm at checking every tile type during test time. Beyond that, things are fuzzy with $RL^2$ and MAML both checking a majority of tile types at test time and E-$RL^2$ being sporadic in this regard. As for the number of times the death tile type was visited, we see that most algorithms start by dying in all three test episodes, and subsequently decrease to between one and two by the time they have converged. As mentioned above, $RL^2$ suffers from finding one goal and exploiting it, whereas the other algorithms regularly explore to find more goals. For the most part, the exploratory algorithms consistently deliver the best performance on these metrics. Performance on the death heuristic and the tile types found heuristic seem to indicate the meta learners are learning how to do at least some system identification.

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

# 7 Appendix A: MAML Rewards Vs Number of Gradient Steps

In the experiment, only 1-step of gradient descent is done in the MAML variants. However, we have explored doing multiple steps of PPO SGD for each task. Example screenshots of multiple gradient update during training are shown below.

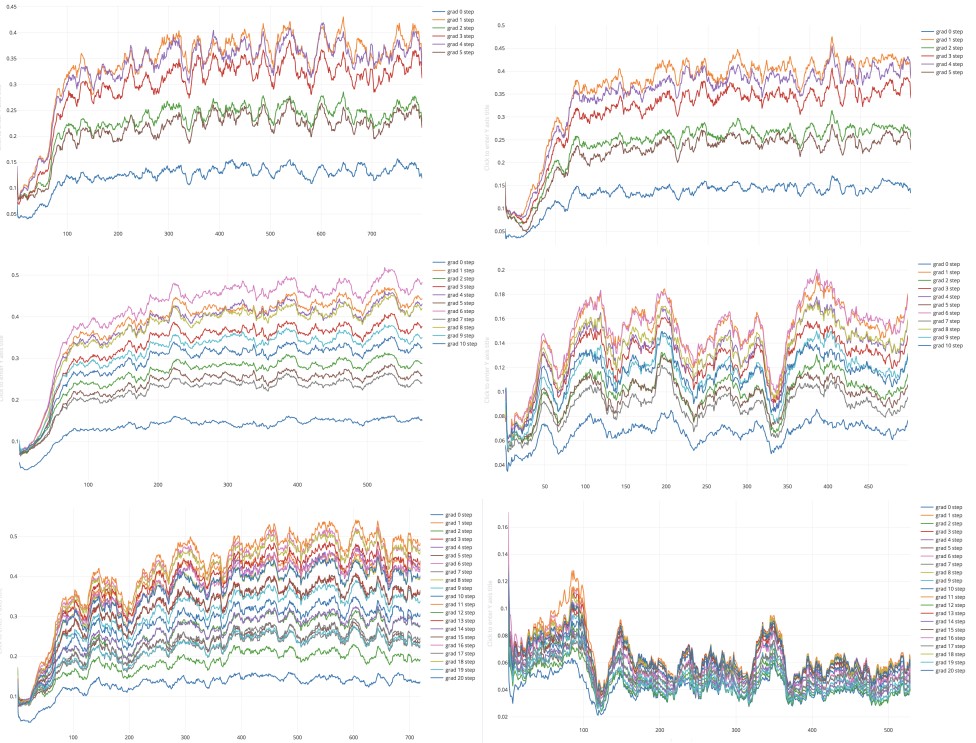

Figure 8: MAML on the right and E-MAML on the left. A look at the number of gradient steps at test time vs reward on the Krazy World environment. Both MAML and E-MAML do not typically benefit from seeing more than one gradient step at test time. Hence, we only perform one gradient step at test time for the experiments in this paper.

## 8   APPENDIX B: E-MAML EXPLORATION AS A CHOICE

The path-integral formulation of the expected reward in E-MAML is

$$\mathcal{L}(\theta_0) = - \iint R(\tau) \cdot \pi(a_i|s_i, U(\theta_0, R(\bar{\tau}), \bar{\tau} \sim \pi(a|\theta_0))) \cdot \pi(\bar{a}_i \,|\bar{s}_i, \theta_0) \, \mathrm{d}\tau \mathrm{d}\bar{\tau}. \tag{9}$$

We can make it easier to tell the role of $U$ by re-writing the action probability $\pi$ and $U$ in type notation where

$$\hat{\pi}(\theta) :\Rightarrow \{(\tau) : \mathbb{R}^+ \Rightarrow \pi(a_t|s_t, \theta)\} \quad \text{and} \quad \hat{U}(\theta_0) : \Theta \Rightarrow U\left(\theta_0, R(\tau), \tau \sim \pi(a|\theta_0)\right), \tag{10}$$

$\hat{\pi}$ returns a function on trajectories $\tau \in \mathcal{T}$, whereas $\hat{U}$ takes in two more variables from it's closure, and returns an updated $\theta \in \Theta$. Using the Polish notation @ applied from-left-to-right, the surrogate loss could be written as

$$\mathcal{L}(\theta_0) = - \iint R(\tau) \cdot \left(\hat{\pi} \circ \hat{U}\right) @\theta_0 @\bar{\tau} \cdot \hat{\pi} @\theta_0 @\bar{\tau} \, \mathrm{d}\tau \mathrm{d}\bar{\tau}. \tag{11}$$

In this notation, it becomes clear that you could pick any arbitrary function for the inner-update operator $\hat{U}$. Under this context, the update operator of choice in E-MAML (and similarly MAML) is an instance from a more general class of exploratory operators $\mathcal{O}$:

$$\hat{U}_{\text{MAML}}(\theta_0) : \Theta \Rightarrow \text{SGD}\left(\mathcal{L}_{\text{PPO}}, \tau(\pi(\theta_0)), \theta_0\right) \tag{12}$$

where the class of operators it belongs to is $\hat{U}_{\text{MAML}} \in \mathcal{O} : \{(\Theta) :\Rightarrow \Theta\}$

So what is this PPO/VPG update operator? One could argue that $U_{\text{MAML}}$ is a "reward-maximizing" exploration strategy. The reward collected during the exploration phase is used to compute the gradient direction for $\theta_0$, which is in-turn used to update $\theta_0 \rightarrow \theta$.

Then it becomes immediately obvious there are a few other operators that could also be plugged in:

- *random exploration* on a single task, where $\theta$ is perturbed randomly.
- *natural gradient* on a single task, where such perturbation is scaled with the inverse fisher information.
- *perpendicular to gradient*: Another extreme is to update the initial parameter $\theta_0$ perpendicular to the gradient direction.
- *ε-greedy*: A middle ground between these extremes is a $\varepsilon$-greedy exploration strategy, where for $\varepsilon$ of the time the per-task exploration is done randomly, whereas for the rest of the time $(1 - \varepsilon)$ the gradient direction is picked.
- *Meta-learned Exploratory Operator* where the inner-update operator $U$ is learned to optimality to a specific class of tasks. See Schmidhuber (2006; 1987).
- *Distraction-free Learning* where the sign of the explorative log probability is flipped. This should discourage exploratory behavior. This could be useful in a life-long learning agent during test time, where exploration is extremely costly.

We are planning on exploring these directions in the future.

