# OpenReview forum: "Some Considerations on Learning to Explore via Meta-Reinforcement Learning"
_ICLR.cc/2018/Conference — Invite to Workshop Track_

### Official Review · AnonReviewer3 · 2017-11-24

**Rating:** 7
**Confidence:** 4

**Review:**

This is an interesting paper about correcting some of the myopic bias in meta RL. For two existing algorithms (MAML, RL2) it proposes a modification of the metaloss that encourages more exploration in the first (couple of) test episodes. The approach is a reasonable one, the proposed methods seem to work, the (toy) domains are appropriate, and the paper is well-rounded with background, motivation and a lot of auxiliary results.

Nevertheless, it could be substantially improved:

Section 4 is of mixed rigor: some aspects are formally defined and clear, others are not defined at all, and in the current state many things are either incomplete or redundant. Please be more rigorous throughout, define all the terms you use (e.g. \tau, R, \bar{\tau}, ...). Actually, the text never makes it clear how \tau and \ber{\tau} relate to each other: make this connection in a formal way, please.

In your (Elman) formulation, “L” is not an RNN, but just a feed-forward mapping?

Equation 3 is over-complicated: it is actually just a product of two integrals, because all the terms are separable.

The integral notation is not meaningful: you can’t sample something in the subscript the way you would in an expectation. Please make this rigorous.

The variability across seems extremely large, so it might be worth averaging over mores seeds for the learning curves, so that differences are more likely to be significant.

Figure fontsizes are too small to read, and the figures in the appendix are impossible to read. Also, I’d recommend always plotting std instead of variance, so that the units or reward remain comparable.

I understand that you built a rich, flexible domain. But please describe the variant you actually use, cleanly, without all the other variants. Or, alternatively, run experiments on multiple variants.

---

> ### Author Response · Authors · 2017-12-24
> **We have fixed the plots and made the notation more clear and rigorous.**
>
> This is an interesting paper about correcting some of the myopic bias in meta RL. For two existing algorithms (MAML, RL2) it proposes a modification of the metaloss that encourages more exploration in the first (couple of) test episodes. The approach is a reasonable one, the proposed methods seem to work, the (toy) domains are appropriate, and the paper is well-rounded with background, motivation and a lot of auxiliary results.
>
> Thank you for this excellent summary and compliment of the work!
>
> =========================================================================
>
> Section 4 is of mixed rigor: some aspects are formally defined and clear, others are not defined at all, and in the current state many things are either incomplete or redundant. Please be more rigorous throughout, define all the terms you use (e.g. \tau, R, \bar{\tau}, ...). Actually, the text never makes it clear how \tau and \ber{\tau} relate to each other: make this connection in a formal way, please.
>
> We have made the suggested improvements, clarifying notation and more explicitly defining tau and \bar{tau}. R was defined in the MDP notation section and means the usual thing for MDPs.
>
> =========================================================================
>
> Equation 3 is over-complicated: it is actually just a product of two integrals, because all the terms are separable.
>
> Yes, this is true. It was not our intention to show off or otherwise make this equation seem more complex than it is. In fact, we were trying to simplify things by not skipping steps and separating the integrals prematurely. We asked our colleagues about this, and the response was mixed with half of them preferring the current notation and the other half preferring earlier separation. If you have strong feelings about this, we are willing to change it for the final version.
> =========================================================================
>
>
> The integral notation is not meaningful: you can’t sample something in the subscript the way you would in an expectation. Please make this rigorous.
>
> This is a fair comment. We were simply trying to make explicit the dependence on the sampling distribution, since it is one of the key insights of our method. However, we agree with you and have changed the notation. We have added an appendix B which seeks to examine some of these choices in a more rigorous context.
>
> =========================================================================
>
>
> The variability across seems extremely large, so it might be worth averaging over mores seeds for the learning curves, so that differences are more likely to be significant.
>
> We did this and it helped substantially with obtaining more smooth results with more significant differences. Thank you for the suggestion it was very helpful!
>
> =========================================================================
>
>
> Figure fontsizes are too small to read, and the figures in the appendix are impossible to read. Also, I’d recommend always plotting std instead of variance, so that the units or reward remain comparable.
>
> Fixed. Thanks!
> =========================================================================
>
>
> I understand that you built a rich, flexible domain. But please describe the variant you actually use, cleanly, without all the other variants. Or, alternatively, run experiments on multiple variants.
>
> We plan to release the source for the domain we used. But the variant we used is the one pictured in the paper, with all options turned on. We can add the environment hyperparameters to an appendix of the paper with a brief description if you think this would be useful.
>
> =========================================================================
>
> Rating: 6: Marginally above acceptance threshold
>
> In light of the fact we have addressed your major concerns with this work, we would appreciate it if you would consider revising your score.

---

> > ### Comment · AnonReviewer3 · 2018-01-02
> > **improved**
> >
> > The revised paper is not perfect, but improved substantially, and addresses multiple issues. I raised my review score.

---

### Official Review · AnonReviewer2 · 2017-11-25
**Interesting direction for exploration in meta-RL. Many relations to prior work missing though. Let's wait for rebuttal.**

**Rating:** 6
**Confidence:** 5

**Review:**

The paper proposes a trick of extending objective functions to drive exploration in meta-RL on top of two recent so-called meta-RL algorithms, Model-Agnostic Meta-Learning (MAML) and RL^2.

Pros:

+ Quite simple but promising idea to augment exploration in MAML and RL^2 by taking initial sampling distribution into account.

+ Excellent analysis of learning curves with variances across two different environments. Charts across different random seeds and hyperparameters indicate  reproducibility.


Cons/Typos/Suggestions:

- The brief introduction to meta-RL is missing lots of related work - see below.

- Equation (3) and equations on the top of page 4: Mathematically, it looks better to swap \mathrm{d}\tau and \mathrm{d}\bar{\tau}, to obtain a consistent ordering with the double integrals.

- In page 4, last paragraph before Section 5, “However, during backward pass, the future discounted returns for the policy gradient computation will zero out the contributions from exploratory episodes”:  I did not fully understand this - please explain better.

- It is not very clear if the authors use REINFORCE or more advanced approaches like TRPO/PPO/DDPG to perform policy gradient updates?

- I'd like to see more detailed hyperparameter settings.

- Figures 10, 11, 12, 13, 14: Too small to see clearly. I would propose to re-arrange the figures in either [2, 2]-layout, or a single column layout, particularly for Figure 14.

- Figures 5, 6, 9: Wouldn't it be better to also use log-scale on the x-axis for consistent comparison with curves in Krazy World experiments ?

3. It could be very interesting to benchmark also in Mujoco environments, such as modified Ant Maze.

Overall, the idea proposed in this paper is interesting. I agree with the authors that a good learner should be able to generalize to new tasks with very few trials compared with learning each task from scratch. This, however, is usually called transfer learning, not metalearning. As mentioned above, experiments in more complex, continuous control tasks with Mujoco simulators might be illuminating.

Relation to prior work:

p 2: Authors write: "Recently, a flurry of new work in Deep Reinforcement Learning has provided the foundations for tackling RL problems that were previously thought intractable. This work includes: 1) Mnih et al. (2015; 2016), which allow for discrete control in complex environments directly from raw images. 2) Schulman et al. (2015); Mnih et al. (2016); Schulman et al. (2017); Lillicrap et al. (2015), which have allowed for high-dimensional continuous control in complex environments from raw state information."

Here it should be mentioned that the first RL for high-dimensional continuous control in complex environments from raw state information was actually published in mid 2013:

(1) Koutnik, J., Cuccu, G., Schmidhuber, J., and Gomez, F. (July 2013). Evolving large-scale neural networks for vision-based reinforcement learning. GECCO 2013, pages 1061-1068, Amsterdam. ACM.

p2: Authors write: "In practice, these methods are often not used due to difficulties with high-dimensional observations, difficulty in implementation on arbitrary domains, and lack of promising results."

Not quite true - RL robots with high-dimensional video inputs and intrinsic motivation learned to explore in 2015:

(2) Kompella, Stollenga, Luciw, Schmidhuber. Continual curiosity-driven skill acquisition from high-dimensional video inputs for humanoid robots. Artificial Intelligence, 2015.

p2: Authors write: "Although this line of work does not explicitly deal with exploration in meta learning, it remains a large source of inspiration for this work."

p2: Authors write: "To the best of our knowledge, there does not exist any literature addressing the topic of exploration in meta RL."

But there is such literature - see the following meta-RL work where exploration is the central issue:

(3) J. Schmidhuber. Exploring the Predictable. In Ghosh, S. Tsutsui, eds., Advances in Evolutionary Computing, p. 579-612, Springer, 2002.

The RL method of this paper is the one from the original meta-RL work:

(4) J. Schmidhuber. On learning how to learn learning strategies. Technical Report FKI-198-94, Fakultät für Informatik, Technische Universität München, November 1994.

Which then led to:

(5) J. Schmidhuber, J.  Zhao, N. Schraudolph. Reinforcement learning with self-modifying policies. In S. Thrun and L. Pratt, eds., Learning to learn, Kluwer, pages 293-309, 1997.

p2: "In hierarchical RL, a major focus is on learning primitives that can be reused and strung together. These primitives will frequently enable better exploration, since they’ll often relate to better coverage over state visitation frequencies. Recent work in this direction includes (Vezhnevets et al., 2017; Bacon & Precup, 2015; Tessler et al., 2016; Rusu et al., 2016)."

These are very recent refs - one should cite original work on hierarchical RL including:

J.  Schmidhuber. Learning to generate sub-goals for action sequences. In T. Kohonen, K. Mäkisara, O. Simula, and J. Kangas, editors, Artificial Neural Networks, pages 967-972. Elsevier Science Publishers B.V., North-Holland, 1991.

M. B. Ring. Incremental Development of Complex Behaviors through Automatic Construction of Sensory-Motor Hierarchies. Machine Learning: Proceedings of the Eighth International Workshop, L. Birnbaum and G. Collins, 343-347, Morgan Kaufmann, 1991.

M. Wiering and J. Schmidhuber. HQ-Learning. Adaptive Behavior 6(2):219-246, 1997

References to original work on meta-RL are missing. How does the approach of the authors relate to the following approaches?

(6) J. Schmidhuber. Gödel machines: Fully Self-Referential Optimal Universal Self-Improvers. In B. Goertzel and C. Pennachin, eds.: Artificial General Intelligence, p. 119-226, 2006.

(7) J. Schmidhuber. Evolutionary principles in self-referential learning, or on learning how to learn: The meta-meta-... hook. Diploma thesis, TUM, 1987.

Papers (4,5) above describe a universal self-referential, self-modifying RL machine. It can implement and run all kinds of learning algorithms on itself, but cannot learn them by gradient descent (because it's RL). Instead it uses what was later called the success-story algorithm (5) to handle all the meta-learning and meta-meta-learning etc.

Ref (7) above also has a universal programming language such that the system can learn to implement and run all kinds of computable learning algorithms, and uses what's now called Genetic Programming (GP), but applied to itself, to recursively evolve better GP methods through meta-GP and meta-meta-GP etc.

Ref (6) is about an optimal way of learning or the initial code of a learning machine through self-modifications, again with a universal programming language such that the system can learn to implement and run all kinds of computable learning algorithms.

General recommendation: Accept, provided the comments are taken into account, and the relation to previous work is established.

---

> ### Author Response · Authors · 2017-12-24
> **We have fixed issues with plots and exposition and addressed the prior literature.**
>
>
> First and foremost, we would like to apologize for having missed the relevant prior work by Schmidhuber et al. We have taken care to better connect our work to this prior work, as detailed below.
>
> =========================================================================
>
> “Equation (3) and equations on the top of page 4: Mathematically, it looks better to swap \mathrm{d}\tau and \mathrm{d}\bar{\tau}, to obtain a consistent ordering with the double integrals.”
>
> Agreed. This change has been made.
>
> =========================================================================
>
>
> “In page 4, last paragraph before Section 5, “However, during backward pass, the future discounted returns for the policy gradient computation will zero out the contributions from exploratory episodes”:  I did not fully understand this - please explain better.”
>
> Please see equation 4 in the latest draft and the accompanying text. We have better explained the procedure.
>
> =========================================================================
>
>
> It is not very clear if the authors use REINFORCE or more advanced approaches like TRPO/PPO/DDPG to perform policy gradient updates?
>
> For E-MAML/MAML, the inner update is VPG and the outer update is PPO. For E-RL2/RL2, PPO is used. We have noted this in the experiments section of the paper.
>
> =========================================================================
>
>
> “I'd like to see more detailed hyperparameter settings.”
> We have included some further discussion on the training procedure in the experiments section. Further, it is our intention to release the code for this paper, which will include the hyper-parameters used in these algorithms. We can also put these hyper-parameters into a table in an appendix of this paper, to ensure redundancy in their availability.
>
> =========================================================================
>
>
> “Figures 10, 11, 12, 13, 14: Too small to see clearly. I would propose to re-arrange the figures in either [2, 2]-layout, or a single column layout, particularly for Figure 14.”
>
> We agree. We have switched to a [2, 2]-layout. The figures are still somewhat small, though when viewed on a computer one can easily zoom in and read them more easily. Of course, we would be willing to move to a single column layout in the final version if the present figures are still too difficult to read.
>
> =========================================================================
>
>
> “Figures 5, 6, 9: Wouldn't it be better to also use log-scale on the x-axis for consistent comparison with curves in Krazy World experiments ?”
>
> We have updated the figures and made the axes consistent.
>
> =========================================================================
>
>
> “It could be very interesting to benchmark also in Mujoco environments, such as modified Ant Maze.”
>
> We have been working on continuous control tasks and would hope to include them in the final version. The difficulties we have thus far encountered with these tasks are interesting, but perhaps outside the scope of this paper at the present time.
>
> =========================================================================
>
>
> “Overall, the idea proposed in this paper is interesting. I agree with the authors that a good learner should be able to generalize to new tasks with very few trials compared with learning each task from scratch. This, however, is usually called transfer learning, not metalearning. As mentioned above, experiments in more complex, continuous control tasks with Mujoco simulators might be illuminating. “
>
> See the above comment regarding continuous control. As for difficulties with terminology, some of this stems from following the leads set in the prior literature (the MAML and RL2 papers) which refer to the problem as meta learning. We have attempted to give a more thorough overview of lifelong learning/transfer learning in this revised draft. Please see our response to the first review for further details.
>
> =========================================================================
>
>
> “(1) Koutnik, J., Cuccu, G., Schmidhuber, J., and Gomez, F. (July 2013). Evolving large-scale neural networks for vision-based reinforcement learning. GECCO 2013, pages 1061-1068, Amsterdam. ACM.”
>
>
> We have added this citation. Apologies for having missed it. This reference was actually in our bib file but for some reason did not make it into the paper proper.

---

> ### Author Response · Authors · 2017-12-24
> **We have fixed issues with plots and exposition and addressed the prior literature Part 2**
>
> =========================================================================
>
>
>
> p2: Authors write: "In practice, these methods are often not used due to difficulties with high-dimensional observations, difficulty in implementation on arbitrary domains, and lack of promising results."
>
> “Not quite true - RL robots with high-dimensional video inputs and intrinsic motivation learned to explore in 2015:
>
> (2) Kompella, Stollenga, Luciw, Schmidhuber. Continual curiosity-driven skill acquisition from high-dimensional video inputs for humanoid robots. Artificial Intelligence, 2015.”
>
>
> We have adjusted the discussion and added this reference.
>
> =========================================================================
>
> p2: Authors write: "Although this line of work does not explicitly deal with exploration in meta learning, it remains a large source of inspiration for this work."
>
> p2: Authors write: "To the best of our knowledge, there does not exist any literature addressing the topic of exploration in meta RL."
>
> “But there is such literature - see the following meta-RL work where exploration is the central issue:
>
> (3) J. Schmidhuber. Exploring the Predictable. In Ghosh, S. Tsutsui, eds., Advances in Evolutionary Computing, p. 579-612, Springer, 2002.”
>
>
> We have adjusted the discussion and added this reference.
>
> =========================================================================
>
>
> “J. Schmidhuber, J.  Zhao, N. Schraudolph. Reinforcement learning with self-modifying policies. In S. Thrun and L. Pratt, eds., Learning to learn, Kluwer, pages 293-309, 1997.”
>
>
> We have added this reference.
>
> =========================================================================

---

> ### Author Response · Authors · 2017-12-24
> **We have fixed issues with plots and exposition and addressed the prior literature Part 3**
>
>
> “p2: "In hierarchical RL, a major focus is on learning primitives that can be reused and strung together. These primitives will frequently enable better exploration, since they’ll often relate to better coverage over state visitation frequencies. Recent work in this direction includes (Vezhnevets et al., 2017; Bacon & Precup, 2015; Tessler et al., 2016; Rusu et al., 2016)."
>
> “These are very recent refs - one should cite original work on hierarchical RL including:
>
> J.  Schmidhuber. Learning to generate sub-goals for action sequences. In T. Kohonen, K. Mäkisara, O. Simula, and J. Kangas, editors, Artificial Neural Networks, pages 967-972. Elsevier Science Publishers B.V., North-Holland, 1991.
>
> M. B. Ring. Incremental Development of Complex Behaviors through Automatic Construction of Sensory-Motor Hierarchies. Machine Learning: Proceedings of the Eighth International Workshop, L. Birnbaum and G. Collins, 343-347, Morgan Kaufmann, 1991.”
>
> M. Wiering and J. Schmidhuber. HQ-Learning. Adaptive Behavior 6(2):219-246, 1997”
>
>
> These refs cite older work in the area, which in turn cites the work you mention. This is not a review paper and hence mentioning every prior work in a field as large as hierarchical RL is not practical nor desired. We have added a review article by Barto and your last reference on HQ learning to account for this.
>
> =========================================================================
>
>
>
>
> “References to original work on meta-RL are missing. How does the approach of the authors relate to the following approaches?
>
> (6) J. Schmidhuber. Gödel machines: Fully Self-Referential Optimal Universal Self-Improvers. In B. Goertzel and C. Pennachin, eds.: Artificial General Intelligence, p. 119-226, 2006.
>
> (7) J. Schmidhuber. Evolutionary principles in self-referential learning, or on learning how to learn: The meta-meta-... hook. Diploma thesis, TUM, 1987.
>
> Papers (4,5) above describe a universal self-referential, self-modifying RL machine. It can implement and run all kinds of learning algorithms on itself, but cannot learn them by gradient descent (because it's RL). Instead it uses what was later called the success-story algorithm (5) to handle all the meta-learning and meta-meta-learning etc.
>
> Ref (7) above also has a universal programming language such that the system can learn to implement and run all kinds of computable learning algorithms, and uses what's now called Genetic Programming (GP), but applied to itself, to recursively evolve better GP methods through meta-GP and meta-meta-GP etc.
>
> Ref (6) is about an optimal way of learning or the initial code of a learning machine through self-modifications, again with a universal programming language such that the system can learn to implement and run all kinds of computable learning algorithms.”
>
> We added several sentences regarding this to our paper. We have also connected this idea to a more broad interpretation of our work. Please see appendix B which cites this work in reference to our algorithm derivation.
> =========================================================================
>
>
> General recommendation: Accept, provided the comments are taken into account, and the relation to previous work is established
>
> We feel the paper now is substantially improved and we exerted significant energy addressing your concerns. Please see in particular the improved figures and heuristic metrics, as well as the improved works cited section, which address the majority of the major issues you had with this work. We would appreciate it if you could reconsider your score in light of these new revisions.
>
>
>
> =========================================================================

---

### Official Review · AnonReviewer1 · 2017-11-27
**A new exploration algorithm for reinforcement learning**

**Rating:** 4
**Confidence:** 4

**Review:**

Summary: this paper proposes algorithmic extensions to two existing RL algorithms to improve exploration in meta-reinforcement learning. The new approach is compared to the baselines on which they are built on a new domain, and a grid-world.

This paper needs substantial revision. The first and primary issue is that authors claim their exists not prior work on "exploration in Meta-RL". This appears to be the case because the authors did not use the usual names for this: life-long learning, learning-to-learn, continual learning, multi-task learning, etc. If you use these terms you see that much of the work in these settings is about how to utilize and adapt exploration. Either given a "free learning phases", exploration based in internal drives (curiosity, intrinsic motivation). These are subfields with too much literature to list here. The paper under-review must survey such literature and discuss why these new approaches are a unique contribution.

The empirical results do not currently support the claimed contributions of the paper. The first batch of results in on a new task introduced by this paper. Why was a new domain introduced? How are existing domains not suitable. This is problematic because domains can easily exhibit designer bias, which is difficult to detect. Designing domains are very difficult and why benchmark domains that have been well vetted by the community are such an important standard. In the experiment, the parameters were randomly sampled---is a very non-conventional choice. Usually one performance a search for the best setting and then compares the results. This would introduce substantial variance in the results, requiring many more runs to make statistically significant conclusions.

The results on the first task are not clear. In fig4 one could argue that e-maml is perhaps performing the best, but the variance of the individual lines makes it difficult to conclude much. In fig5 rl2 gets the best final performance---do you have a hypothesis as to why? Much more analysis of the results is needed.

There are well-known measures used in transfer learning to access performance, such as jump-start. Why did you define new ones here?

Figure 6 is difficult to read. Why not define the Gap and then plot the gap. These are very unclear plots especially bottom right. It's your job to sub-select and highlight results to clearly support the contribution of the paper---that is not the case here. Same thing with figure 7. I am not sure what to conclude from this graph.

The paper, overall is very informal and unpolished. The text is littered with colloquial language, which though fun, is not as precise as required for technical documents. Meta-RL is never formally and precisely defined. There are many strong statements e.g., : "which indicates that at the very least the meta learning is able to do system identification correctly.">> none of the results support such a claim. Expectations and policies are defined with U which is never formally defined. The background states the problem of study is a finite horizon MDP, but I think they mean episodic tasks. The word heuristic is used, when really should be metric or measure.

---

> ### Author Response · Authors · 2017-12-24
> **We have added discussion of prior literature and better highlighted the novelty of our contributions.**
>
> The first and primary issue is that authors claim their exists not prior work on "exploration in Meta-RL"....The paper under-review must survey such literature and discuss why these new approaches are a unique contribution.
>
> We have added numerous references to these fields in the related literature section of the paper and clarified our contribution in this context. We are interested in the problem of meta-learning for RL (which largely deals with finding initializations that are quick to adapt to new domains). This problem ends up having a different formulation from the areas mentioned above. Our specific contribution is the creation of two new algorithms that find good initializations for RL algorithms to quickly adapt to new domains, yet do not sacrifice exploratory power to obtain these initializations. We show further that one can consider a large number of interesting algorithms for finding initializations that are good at exploring. This is also a novel contribution.
> =========================================================================
>
>
> The empirical results do not currently support the claimed contributions of the paper.
>
> The results have been strengthened since the initial submission. It is now clear that our methods provide substantially better performance. Further, the heuristic metrics indicate they are superior at exploration.
>
> =========================================================================
>
> The first batch of results in on a new task introduced by this paper. Why was a new domain introduced? How are existing domains not suitable.
>
> The domains are gridworlds and mazes, neither of which should require this sort of justification prior to use. The gridworld does not use a standard reference implementation (we am not aware of any such implementation) and was designed so that its level of difficulty could be more easily controlled during experimentation.
>
> =========================================================================
>
> Designing domains are very difficult and why benchmark domains that have been well vetted by the community are such an important standard
> We agree with this. And indeed, we ourselves have designed reference domains for RL problems that are extremely popular in the community. In these cases, the domains were usually derived from an initial paper such as this one and subsequently improved upon by the community over time. In our experience, releasing a new domain in the context of this paper aligns well with how our previous successful domains have been released.
> =========================================================================
>
> In the experiment, the parameters were randomly sampled---is a very non-conventional choice. Usually one performance a search for the best setting and then compares the results. This would introduce substantial variance in the results, requiring many more runs to make statistically significant conclusions.
>
> We have averaged over many more trials and this has significantly smoothed the curves. We were trying to avoid overfitting, which is a systematic problem in the way RL results are typically reported.
>
> =========================================================================
>
>
> The results on the first task are not clear. In fig4 one could argue that e-maml is perhaps performing the best, but the variance of the individual lines makes it difficult to conclude much. In fig5 rl2 gets the best final performance---do you have a hypothesis as to why? Much more analysis of the results is needed.
>
> The result are more clear now and RL2 no longer gets the best final performance. Also, an important thing to consider is how fast the algorithms approach their final performance. For instance, in the referenced graph, E-MAML converged within ~10 million timesteps whereas RL2 took nearly twice as long. We apologize for not making this important point more explicit in the paper. In any case, this particular comment has been outmoded.
>
> =========================================================================
>
>
> There are well-known measures used in transfer learning to access performance, such as jump-start. Why did you define new ones here?
>
> Jump start is quite similar to the gap metric we consider in the paper. We have clarified this.
>
> =========================================================================

---

> ### Author Response · Authors · 2017-12-24
> **We have added discussion of prior literature and better highlighted the novelty of our contributions Part 2**
>
>
> Figure 6 is difficult to read.
>
> The figures have been dramatically improved. We apologize for the poor initial pass.
>
> =========================================================================
>
>
> Why not define the Gap and then plot the gap.
>
> We feel it is illustrative to see the initial policy and the post-update policy in the same place. Actually seeing the gap between the two algorithms can be easier to interpret than the gap itself, which is a scalar.
>
> =========================================================================
>
>
> These are very unclear plots especially bottom right. It's your job to sub-select and highlight results to clearly support the contribution of the paper---that is not the case here. Same thing with figure 7. I am not sure what to conclude from this graph.
>
> We took these comments to heart and exerted a lot of effort on improving the plots. We solicited feedback from our colleagues who suggest the new plots are much more clear, readable, and better convey our points. We also took better care to clarify this in our captions.
>
> =========================================================================
>
> The paper, overall is very informal and unpolished. The text is littered with colloquial language, which though fun, is not as precise as required for technical documents. Meta-RL is never formally and precisely defined. There are many strong statements e.g., : "which indicates that at the very least the meta learning is able to do system identification correctly.">> none of the results support such a claim. Expectations and policies are defined with U which is never formally defined. The background states the problem of study is a finite horizon MDP, but I think they mean episodic tasks. The word heuristic is used, when really should be metric or measure.
>
> Thank you for these comments. We have cleaned up the writing.
> =========================================================================

---

### Author Response · Authors · 2017-12-24
**We have made substantial changes based on reviewer comments**

The following concerns were listed across multiple reviewers:

1) Our paper misses citations wherein the similar problems are considered under different names. This problem is quite a large one, and it is unfortunate that the literature is at times disjoint and difficult to search. You will notice that the first and second reviewer both told us that we missed many essential references, but the crucial missed references provided by both are entirely different. Further, the third reviewer did not indicate any issues with the literature we cited. We believe this indicates the difficulty in accurately capturing prior work in this area.

2) The graphs suffered from a variety of deficiencies. These deficiencies were both major (not clearly and convincingly demonstrating the strengths of our proposed methods) and minor (the text or graphs themselves being at times too small).

3) There were portions of the paper that appeared hastily written or wherein spelling and grammatical mistakes were present. Further, there were claims that the reviewers felt were not sufficiently substantiated and parts of the paper lacked rigor.

We have addressed these concerns in the following ways:

1) We have made an effort to address relevant prior literature. In particular, we have better explained the work’s connection to prior work by Schmidhuber et al and better explained what distinguishes this work from prior work on lifelong learning. See responses to individual reviewers for a more thorough explanation of these changes. Further, we have included an additional appendix which highlights our algorithmic development as a novel process for investigating exploration in meta-RL. We feel this appendix should completely remove any doubts regarding the novelty of this work.

2) As for the graphs, we have fixed the presentation and layout issues. We have averaged over more seeds, which decreased the overall reported standard deviation across all algorithms, thus making the graphs more legible. We have also separated the learning curves onto multiple plots so that we can directly plot the standard deviations onto the learning curves without the plots appearing too busy.

3) We have carefully edited the paper and fixed any substandard writing. We have also taken care to properly define notation, and made several improvements to the notation. We improved the writing’s clarity, and better highlighted the strength of our contributions. We removed several claims that the reviewers felt were too strong, and replaced them with more agreeable claims that are better supported by the experimental results. We have added an interesting new appendix which considers some of our insights in a more formal and rigorous manner. Finally, we have completely rewritten the experiments section, better explaining the experimental procedure.


Please see the responses to individual reviews below for further elaboration on specific changes we made to address reviewer comments.

---

### Decision · Program_Chairs · 2018-01-29
**ICLR 2018 Conference Acceptance Decision**

**Decision:**

Invite to Workshop Track

**Comment:**

Overall, the paper is missing a couple of ingredients that would put it over the bar for acceptance:

- I am mystified by statements such as "RL2 no longer gets the best final performance." from one revision to another, as I have lower confidence in the results now.

- More importantly, the paper is missing comparisons of the proposed methods on *already existing* benchmarks. I agree with Reviewer 1 that a paper that only compares on benchmarks introduced in the very same submission is not as strong as it could be.

In general, the idea seems interesting and compelling enough (at least on the Krazy World & maze environments) that I can recommend inviting to the workshop track.